# Segment-Based Unsupervised Learning Method in Sensor-Based Human Activity Recognition

**DOI:** 10.3390/s23208449

**Published:** 2023-10-13

**Authors:** Koki Takenaka, Kei Kondo, Tatsuhito Hasegawa

**Affiliations:** Graduate School of Engineering, University of Fukui, Fukui 910-8507, Japan; kicont115sub@gmail.com (K.K.); t-hase@u-fukui.ac.jp (T.H.)

**Keywords:** human activity recognition, unsupervised representation learning, accelerometer sensor data, segment data

## Abstract

Sensor-based human activity recognition (HAR) is a task to recognize human activities, and HAR has an important role in analyzing human behavior such as in the healthcare field. HAR is typically implemented using traditional machine learning methods. In contrast to traditional machine learning methods, deep learning models can be trained end-to-end with automatic feature extraction from raw sensor data. Therefore, deep learning models can adapt to various situations. However, deep learning models require substantial amounts of training data, and annotating activity labels to construct a training dataset is cost-intensive due to the need for human labor. In this study, we focused on the continuity of activities and propose a segment-based unsupervised deep learning method for HAR using accelerometer sensor data. We define segment data as sensor data measured at one time, and this includes only a single activity. To collect the segment data, we propose a measurement method where the users only need to annotate the starting, changing, and ending points of their activity rather than the activity label. We developed a new segment-based SimCLR, which uses pairs of segment data, and propose a method that combines segment-based SimCLR with SDFD. We investigated the effectiveness of feature representations obtained by training the linear layer with fixed weights obtained by unsupervised learning methods. As a result, we demonstrated that the proposed combined method acquires generalized feature representations. The results of transfer learning on different datasets suggest that the proposed method is robust to the sampling frequency of the sensor data, although it requires more training data than other methods.

## 1. Introduction

Human activity recognition (HAR) using sensor data is a task that recognizes human activities wherein the data are measured by sensors such as accelerometers and gyroscopes. For example, human activities such as “walking” and “running” were predicted from the measured accelerometer data. This technology is used in various applications from analyzing sports movements [1] to healthcare such as health awareness maintenance and the detection of risky movements by patients [2,3,4]. HAR is an essential technology because the predicted results have some influence on decision-making.

Recently, neural networks (NNs) have been used for HAR [5,6,7,8]. HAR is typically implemented using machine learning methods [9,10] such as support vector machines (SVMs) and hidden Markov models (HMMs). Machine learning models require a training dataset consisting of handcrafted features extracted from sensor data and corresponding activity labels. In contrast to traditional machine learning, deep learning methods such as NNs have the advantage of automatic feature extraction, which enables end-to-end training and prediction from raw sensor data. Therefore, deep learning models can adapt to various situations [11] without requiring domain expertise for feature extraction. NNs require substantial amounts of training data to acquire generalized feature representations from sensor data compared to traditional methods. In recent years, the widespread use of smartphones and wearable devices has made it easy to measure sensor data. However, the annotation process is time-consuming and labor-intensive to generate a large amount of training data. Thus, there is a need for an unsupervised learning method for NNs.

In the case of sensor-based HAR, we can design how to measure the sensor data for the training dataset. For example, there are two measurement methods for HAR: the sequential measurement of multiple activities and the individual measurement of each activity. When each activity is measured individually without annotation, the accurate activity of the measurement data is unclear (annotator needs to determine the activity label using video, etc.) However, it can be understood that each of the data corresponds to a single activity.

Our purpose was to develop an unsupervised learning method that leverages characteristics implicitly possessed by sensor data for sensor-based HAR using deep learning methods. Existing unsupervised learning techniques for time series data are based on time series characteristics [12,13] or masks [14], but these methods do not focus on the structure of the training data. Xiao et al. [15] proposed a method employing contrastive loss. This approach calculates the loss in the intermediate layers, and it is not entirely independent of the model’s structure. Franceschi et al. [16] introduced triplet loss (T-Loss), which selects positive samples from the same subseries as a given time series and randomly selects negative samples from other subseries. When utilizing segmented data, this selection method involves considering the data structure, where positive samples should be chosen from the same segment and negative samples should be chosen from different segments. SimCLR employs the normalized temperature-scaled cross-entropy (NT-Xent) loss [17,18], an expanded loss function that is derived from the triplet loss by incorporating multiple negative samples. As this loss function is compatible with the approach of Franceschi et al. [16], there is room for improvement in utilizing segmented data. We investigated a method that focuses on the data structure for unsupervised learning with discriminators [19]. In this study, we introduced SimCLR, which performs contrastive feature representation based on segments, to segment discrimination and feature decorrelation (SDFD) [19]. We combined these methods to develop a sensor-based unsupervised HAR learning method that does not rely on model structure, but focuses on the structure of segmented data. We refer to segment-based SimCLR as SimCLR (seg) and the combined method segment-based SimCLR with SDFD as SimCLR (seg) + SDFD. In our experiments, in order to evaluate the generalization performance of the proposed method, we conducted a linear evaluation under the two task settings: transfer learning within the same domain and transfer learning across different domains.

In summary, our contributions are as follows:We propose a new unsupervised learning method, SimCLR (seg) + SDFD, for sensor-based HAR in an environment where the training dataset consists of segmented data. In the proposed method, segmented data are used only during training and are not used during the evaluation.The ablation study showed that SimCLR (seg) and SDFD improved the f1-score compared to methods that do not focus on the data structure. This shows that focusing on the structure of the training dataset is effective.We demonstrated that our proposed method acquires generalized feature representations for sensor-based HAR through our experiments.

## 2. Related Works

In sensor-based HAR, models are typically trained using data consisting of sensor data and corresponding activity labels. Liu et al. [10] and Hartmann et al. [20] worked on real-time HAR using HMMs, and Liu et al. [21] demonstrated that HMMs are computationally efficient and fully adaptable to real-time applications. Bento et al. [22] demonstrated that traditional machine learning methods using handcrafted features of sensor data can generalize to data from different domains than the training data. These traditional methods require manual feature extraction, whereas deep learning methods extract features automatically. Thus, deep learning methods can be trained and predicted end-to-end and have demonstrated their adaptive capabilities [11].

For sensor-based HAR using deep learning methods, various methods using convolutional NNs (CNNs) [5,6,7,8] and transformers [23] are proposed. Mahmud et al. [24] proposed a method designed to acquire useful features for HAR from different perspectives by applying multiple transformation methods to sensor data. Sanchez et al. [8] proposed a method that converts sensor data into an image format for HAR. These studies focused on an input format for sensor data to improve recognition accuracy. Approaches to improve the architecture of NNs for HAR have been proposed, such as a method that uses multiple models with different roles [25] and another that uses attention mechanisms [26]. Some studies tackle representation learning with activity labels [27,28]. These studies focus on acquiring feature representations for HAR using datasets with activity labels. Deep learning models require more training data for automatic feature extraction [24,29]. In recent years, the widespread use of smartphones and wearable devices has enabled the collection of large amounts of sensor data. However, annotators need to manually annotate activity labels for each of the measurement data, which requires more-significant human resources and time. On the other hand, this study focuses on learning methods without activity labels, which in turn reduces annotation costs.

In image recognition, several unsupervised learning methods have been proposed [30,31]. Tao et al. [32] proposed a clustering-friendly representation learning method by utilizing instance discrimination (ID) [33], which classifies each of the data as a unique class and removes the redundant correlation of features. Chen et al. [18] proposed SimCLR as a contrastive learning method that acquires feature representations by a straightforward method. Grill et al. [34] proposed bootstrap your own latent (BYOL) as an efficient representation learning method without using negative examples. Since these methods do not depend on a model architecture, they can be applied to unsupervised learning methods with sensor data by changing an encoder architecture into layers for sensor data.

Methods for segmenting unlabeled sensor data have been proposed. Rodrigues et al. [35] proposed a method for detecting the similarity between continuous time series data. Folgado et al. [36] utilized query sequences for data segmentation. These methods require continuous measurement data or query sequences to compute the similarity and features of time series data. Our approach, on the other hand, allows the model trained with our method to compute features from short-term measurement data without the need to retain training data during inference. Furthermore, these methods focus on unsupervised segmentation of time series data. Our approach focuses on training with segmented unsupervised data, and the trained model can be applied to other classification tasks using short-term data.

Moreover, unsupervised learning methods for HAR using deep learning methods have been proposed to acquire feature representations from unlabeled sensor data [16,29]. Focusing on time series, Haresamudram et al. [12] proposed a method to use contrastive predictive coding [37] for HAR. The method trains a model by comparing a feature map obtained from sensor data with a predicted feature map. Ma et al. [29] proposed a loss function that uses a reconstruction error, a neighborhood of manually designed features, and a neighborhood of time series data to use an auto-encoder (AE). Tonekaboni et al. [13] introduced the concept of a neighborhood with stationary time series data to propose a neighborhood-based unsupervised learning method for non-stationary multivariate time series data. These methods utilize the neighborhood of time series data. We also used the neighborhood of time series data as the segment data to acquire the feature representations for HAR.

## 3. Proposed Method

### 3.1. Data-Measurement Method

Figure 1a,b show two typical measurement methods for HAR. The prior-labeling type in Figure 1a shows that an activity is labeled before the subject initiates an activity. For example, to start the walking measurement, the subject presses the “walk” button. Then, to start the running measurement, the subject presses the “run” button. Therefore, in the prior-labeling type (Figure 1a), subjects need to select the activity label before the measurement each time. The post-labeling type in Figure 1b shows that an activity is labeled after the subject initiates an activity. For example, the subjects move freely with the attached sensor while his/her behavior is recorded by video. After the measurement, the recorded video is visually checked, and each activity is labeled at the given timestamp.

These methods have advantages and disadvantages. Comparing the prior-labeling type (Figure 1a) and the post-labeling type (Figure 1b), the prior-labeling type requires subjects to select an activity label before and after the measurement and is measured for each activity. On the other hand, the post-labeling type does not require the selection of an activity label. In addition, the post-labeling type can continuously measure a series of activities, which reduces the amount of work before and after the measurement, thus reducing the subject’s burden compared to the prior-labeling type. In the prior-labeling type, an annotator does not need to perform any work after the measurement because the activity label is selected before the measurement. In the post-labeling type, the annotator must manually assign the activity label corresponding to the measurement data. As specific examples, Kwapisz et al. [38] and Zhang et al. [39] used the prior-labeling type: Ichino et al. [40,41] and Chavarriaga et al. [42] used the post-labeling type for measurement.

The aim of our method was to acquire feature representations of single activities. The single activity is a motion that consists of a single movement, such as jumping, sitting, or lying down, or a cycle of several consecutive movements, such as walking, running, or climbing stairs [43]. We call data that record one continuous single activity a segment. Our method does not require the annotation of the segment data.

In our proposed method, we used a dataset that can be created with the unlabeled segmented measurement type shown in Figure 1c. This measurement method enables automatic segmentation of measurement data and eliminates the need for annotation work from the measurement methods used in related studies [44,45]. Compared to the two measurement types, the unlabeled segmented measurement type only requires the subject to indicate the start and end of the measurement, which can be any activity in the segmented data.

### 3.2. Preprocessing

We utilized unlabeled segment data for unsupervised learning. We used benchmark datasets in which the sensor data are stored as segment data. In the case of a benchmark dataset that does not have stored segment data, we need to create segment data using segmentation methods [35,36].

A training dataset *D* consists of input data created from segment data and pseudo-label pairs. The input data are instances of segment data divided by a sliding window method with a window size *L* and a stride width *S*. We show the creation method of pseudo-labels in Figure 2. The pseudo-labels are unique values assigned to the segment data and are called segment labels. On the other hand, pseudo-labels assigned to instance data are called instance labels. A total number of instance labels and segment labels, donated by *N* and *M*, respectively, has a relation M≤N.

### 3.3. Model Training Method

Figure 3 shows an overview of the proposed method utilizing segment data. In the proposed method, the model consists of an encoder $f_{e}$ and two projectors $f_{p1},f_{p2}$. The encoder is used to acquire a feature representation of segment data, and the projectors are used for segment label prediction and contrastive loss calculation. We updated the model parameters by using loss functions in SimCLR and SDFD.

#### 3.3.1. Segment Discrimination

Segment discrimination (SD) is an ID-based method that differs from ID in the assignment of pseudo-labels. ID is an unsupervised learning method proposed by Wu et al. [33]. The SD method uses pseudo-labels like ID, but it uses segment labels instead of instance labels, as shown in Figure 2. In the SD implementation, the memory bank [33] size is the same as the number of segment labels. The memory bank is a set of normalized feature representations. Meanwhile, for SD, a model is trained in the same manner as ID using a segment label instead of an instance label. When the number of segments used in SD is *M*, the memory bank is $\left\{f_{1},f_{2},\cdots ,f_{M}\right\}$. When the feature $f=f_{p2}\left(f_{e}\left(x\right)\right)$ obtained from the input data x and the segment label is *s*, the probability P(s|f) is expressed as follows: (1)$$P\left(s|f\right)=\frac{\exp(f_{s}^{⊤}f)}{\sum_{j=1}^{S}\exp\left(f_{j}^{⊤}f\right)}.$$

In SD, it is known that the *i*-th input data $x_{i}$ is the $s_{i}$-th segment. Therefore, the loss function $L_{sd}$ using the feature in the memory bank is
(2)$$L_{sd}=-\sum_{i=1}^{N}\log P\left(s_{i}|f_{i}\right)=-\sum_{i=1}^{N}\log\frac{\exp(f_{s_{i}}^{⊤}f)}{\sum_{j=1}^{S}\exp\left(f_{j}^{⊤}f\right)}.$$

There are two advantages to using SD. The first is that the number of output dimensions can be fixed. The total number of segment labels is less than the total number of instance labels. However, compared to the output size of a typical classification model, it can be extremely large. SD is solved in the same way as ID. Second, it is possible to utilize information implicit in the segments. The assigned segment labels allow input data based on the segment data to output the same features as the same activity. This is expected to be robust to phase differences between the input data.

#### 3.3.2. SD and Feature Decorrelation

SD and feature decorrelation (SDFD) is a method using SD for IDFD as proposed by Tao et al. [32]. The loss function $L_{idfd}$ used in IDFD is a combination of the loss function $L_{id}$ used in ID and a loss function $L_{fd}$ called feature-independent softmax (FIS). Since SDFD is based on SD, the loss function of SDFD is $L_{sdfd}=L_{sd}+L_{fd}$, which changes the ID loss function part from the IDFD loss function $L_{idfd}=L_{id}+L_{fd}$.

FIS is a soft orthogonal constraint on the feature representation that allows non-zeros. Tao et al. [32] demonstrated the learning stability advantage of using FIS over an ordinary orthogonal constraint. Let us denote the feature representation for each batch $F=[f_{1},f_{2},\cdots ,f_{b}]$ as the dimension of the feature representation *d*, the batch size *b*, and the *i*-th feature representation $f_{i}$. $V=F^{⊤}=\left[v_{1},v_{2},\cdots ,v_{d}\right]$ is used, where the FIS loss function $L_{fd}$ is as follows: (3)$$L_{fd}=-\sum_{i=1}^{d}\log\frac{\exp(v_{i}^{⊤}v_{i})}{\sum_{j=1}^{d}\exp\left(v_{j}^{⊤}v_{i}\right)}.$$

We expect that the use of FIS in IDFD and SDFD will make the elements of the feature representation non-correlated.

#### 3.3.3. Segment-Based SimCLR

SimCLR (seg) is the segment-based SimCLR, which uses a segment to create positive pairs. Figure 4 shows the difference in how positive pairs are created. When SimCLR is applied to HAR, the input positive pairs are created from instance data divided by the window size. In contrast, SimCLR (seg) positive pairs are created by selecting two instance data from the same segment.

In SimCLR (seg), the loss function is the same as the loss function used in SimCLR. A positive pair consists of two instance data, $x_{i}$ and $x_{j}$, which is created from the same segment. Let *t* and $t^{\prime }$ be two data augmentation methods and $f\left(x\right)=f_{p1}\left(f_{e}\left(x\right)\right)$ be the model to be trained. The feature representations of the two positive pairs used in the SimCLR loss function are $z_{i}=f\left(t\left(x_{i}\right)\right)$ and $z_{j}=f\left(t^{\prime }\left(x_{j}\right)\right)$. Therefore, the SimCLR loss function $L_{s}$ is expressed using the feature representations of the positive pairs $z_{i}$ and $z_{j}$ as
(4)$$L_{s}=-\log\frac{\exp\left(\frac{\mathrm{sim}(z_{i},z_{j})}{\tau }\right)}{\sum_{k=1}^{2b}1\left(k\neq i\right)\exp\left(\frac{\mathrm{sim}(z_{i},z_{k})}{\tau }\right)}.$$

Here, τ is the temperature parameter, *b* is the batch size, 1 is the indicator function, and sim is the cosine similarity. We used τ=1 for simple cross-entropy in this study. Feature representations are expected to be acquired in a measured activity by creating positive pairs based on segments.

#### 3.3.4. Segment-Based SimCLR with SDFD

We introduce a method that combines SDFD and SimCLR (seg). The input data are created in the same manner as SimCLR (seg). Since an SDFD loss function does not require a data pair when passing the data to the loss function, the pair is combined and passed as one batch. When combined, the loss function *L* is: (5)$$L=\lambda_{1}L_{s}+\lambda_{2}L_{sd}+\lambda_{3}L_{fd}.$$

In this loss function, $\lambda_{1}$, $\lambda_{2}$, and $\lambda_{3}$ denote the weights of each respective loss function. We set $\lambda_{2}=\lambda_{3}=1$ following the work [32] and set $\lambda_{1}$ to 1. Since tuning the hyperparameters is time-consuming [46,47], it may be possible to improve accuracy by adjusting these parameters.

We discuss the computational complexity of the loss function of the proposed method. The computational complexity of the loss function for SimCLR (seg) is the same as that of SimCLR, since the only change is in the data-selection method. When changing from IDFD to SDFD, the computational complexity per batch is reduced by 𝒪(bd(N−M)) because the memory bank is treated as a matrix. Since SimCLR outputs twice as many feature maps as the batch size in a batch, the computational complexity of the proposed method is the sum of the computational complexity of SimCLR and twice the computational complexity of SDFD.

## 4. Experiments

### 4.1. Experimental Setup

We evaluated the proposed method on five benchmark datasets for HAR (HASC [40], Wisdm [38], Opportunity [42], UCI-HAR [48], and USC-HAD [39]). We preprocessed these benchmark datasets into datasets composed of input data and corresponding segment labels. Because the HASC [40] dataset contains multiple sensor devices, we used only measurement data with a sampling frequency of 100 Hz from Apple devices in this experiment. UCI-HAR [48] was divided in advance with a window size of 120 and a stride width of 60. In order to apply the proposed method, segment labels are required. Therefore, UCI-HAR [48] segment information was inferred by comparing the data to create segment labels. In this study, it was assumed that the change point ground truth information is known in order to focus on segment utilization effectiveness. We used only accelerometer data since this is common to each dataset.

We compared the proposed method and baseline methods with a shallow CNN and MLP as a backbone model. The baseline methods were ID [33], IDFD [32], SimCLR [18], and BYOL [34]. ID [33] and IDFD [32] are self-supervised learning methods that classify instance feature representations, with IDFD [32] incorporating an approach where feature representations become uncorrelated from the ID [33]. SimCLR [18] and BYOL [34] are contrastive learning methods that perform model training by comparing feature representations. While SimCLR [18] requires positive and negative pairs to train, BYOL [34] is an extension of SimCLR to allow training with positive pairs only. For the backbones of these methods, we used a three-layer CNN as an encoder and a two-layer multi-layer perceptron (MLP) for the projectors. In the case of BYOL [34], the projector and predictor had a BatchNorm1d in two-layer MLPs. We set the number of output dimensions of the encoder to 128 and the number of input and output dimensions of all projectors and predictors to 128.

We investigated the generalization performance of feature representations acquired in few-shot transfer learning (TL) scenarios. In our experiments, we initially constructed a dataset consisting of pairs of segment labels and instance data for training, validation, and testing from benchmark datasets. We split the training, validation, and testing datasets by randomly selecting subjects for each dataset. Table 1 shows the number of subjects in each dataset. We then trained an encoder using an unsupervised learning method with the training dataset as a pretext task. Finally, we performed TL by selecting a subset of the training dataset. We assessed the generalization performance of the feature representations by comparing the f1-score when classified. During TL, we fixed the weight of the encoder and trained only the linear layer attached to the end of the encoder.

We used a partially different training setup for the pretext task and TL. For the pretext task, we set a batch size of 64 for Opportunity [42] and Wisdm [38] and a batch size of 256 for the other datasets. We used an optimization function, Adam, with a learning rate of 0.01 and 150 epochs. As data augmentations [49], we applied jitter, scaling, and permutation for UCI-HAR [48] and USC-HAD [39] and jitter, scaling, and rotate for the other datasets. We used the weights that resulted in the lowest learning loss during training. For TL, we used a batch size of 512 for UCI-HAR [48] and HASC [40] and 256 for the other datasets. We trained the linear layer with a learning rate of 0.001 and 300 epochs without data augmentations. We used the weights with the lowest validation loss for the evaluation. We set the other settings the same as for the pretext task.

The window size selected has room for discussion. Based on the selected window size and the sampling frequency of the respective dataset, the duration for the utilized instances ranged from a minimum of 2.56 s to a maximum of 6.4 s. Specifically, for the HASC dataset, each instance had a duration of 2.56 s. In addition, it is highly probable that at least a single movement is included, considering the fact that the single movements are at intervals of 1–2 s [43]. Therefore, the experimental environment was considered to have no lack of activity features in a single instance. We confirmed that the proposed method was able to acquire a feature representation of the activity in at least such an environment.

### 4.2. Ten-Shot Transfer Learning Verification

We tested the effectiveness of the feature representation acquired in the pretext task by performing transfer learning (TL) with 10-shot data per class. In this experiment, we randomly selected 10-shot data from each activity in the training dataset and trained the linear layer by TL. Table 2 shows the average f1-scores for five trials. The Non-pre in Table 2 shows the results of training the entire model with initial weights on the selected 10-shots without a pretext task. The proposed SimCLR (seg) + SDFD achieved 70.12%, 76.36%, and 91.08% on the HASC, Wisdm, and UCI-HAR, respectively. In particular, the HASC results showed a 7.26% improvement in the f1-score compared to SimCLR, which does not focus on segments. We considered that the reason for the relatively high score of Non-pre in the results of Wisdm and UCI-HAR is that we used a small model and the training dataset was sufficient to train the entire model for classification. These results showed that the proposed method can effectively acquire feature representations in unsupervised learning methods using sensor data.

Table 3 shows the results of the ablation study. Comparing IDFD and SDFD, the Wisdm results showed an f1-score improvement of 10.18%. The SimCLR (seg) results showed an improvement from the SimCLR results by 6.62%, 9.62%, and 4.94% for HASC, Wisdm, and UCI-HAR, respectively. The results for SDFD and SimCLR (seg) showed that SimCLR (seg) was better for HASC and SDFD was better for Wisdm and UCI-HAR. SimCLR (seg) + SDFD outperformed SimCLR (seg) and SDFD by 0.64% and 0.61% for HASC and UCI-HAR, respectively. SimCLR (seg) + SDFD was 0.43% lower than SimCLR (seg) in Wisdm. The results showed that introducing the segment-based method improved the f1-score. Considering the training dataset size, SimCLR (seg) requires a larger number of subjects than SDFD to acquire feature representations, but its combination with SDFD may have made it possible to acquire feature representations with a relatively small number of subjects. Considering the effectiveness of FIS and segment-based approaches, we considered it important to group similar features in unsupervised HAR in order to acquire more-generalized feature representations.

### 4.3. Confusion Matrix

Figure 5 shows the confusion matrix resulting from TL using 10-shot data for each class in HASC. We presented the confusion matrix using the most-accurate results from the five trials. The HASC dataset contains six activities: “stay”, “walk”, “jog”, “skip”, “stUp” (stair up), and “stDown” (stair down). The confusion matrix revealed that “walk”, “stUp”, and “stDown” were frequently misclassified. Compared to these results, the proposed method made more errors for “stUp”, but improved the prediction of “walk” and “stDown”. Table 4 shows the average f1-scores for each activity. The results indicate that the proposed method had higher scores for “walk”, “jog”, “skip”, and “stDown”, but lower scores for “stay” and “stUp” compared to other methods. SDFD and SimCLR (seg) showed similar or higher scores for most activities compared to IDFD and SimCLR, respectively. The proposed method achieved the highest f1-scores of 87.38% and 82.68% for “jog” and “skip”, respectively. The proposed method improved the scores of “walk” and “stDown” by up to 16.23% and 11.93%, respectively, and decreased the scores of “stUp” by 1.90% compared to IDFD and SimCLR. From these results, we considered that the proposed method achieved a better separation of the “walk” and “stDown” feature representations, but brought the “walk” and “stUp” feature representations closer together. Furthermore, based on the confusion matrix results, it can be considered that the proposed method mitigated the bias in predicting “stUp”.

### 4.4. Instance Effectiveness for Few-Shot TL

We investigated the effect of the number of instances in few-shot TL. We compared the average f1-scores of five trials by performing TL on randomly selected few-shot data per class. Figure 6 shows the average f1-scores for five trials when only the linear layer was trained. The legend represents methods with pretext tasks, and Non-pre represents a method that trained the entire model without pretext tasks. All methods tended to have higher f1-score as the number of instances increased. When the number of instances was five and ten, Non-pre showed low values at 33.41% and 37.14%, respectively. Compared to Non-pre, the models trained with pre-training showed high f1-scores. In particular, SimCLR (seg) and SimCLR (seg) + SDFD showed relatively high f1-scores at 62.02% and 60.02%, respectively, for five-shot TL. As the number of labels increased, the f1-score of SDFD and SimCLR (seg) + SDFD also increased. SimCLR (seg) and SDFD showed relatively high f1-scores when the number of labeled data was 5 and 100, respectively.

The combination of SimCLR (seg) and SDFD improved the f1-score of SDFD in TL with extremely small numbers of labels.

### 4.5. Effectiveness for Different Domains

We investigated whether the proposed method can acquire a generalized feature representation to different domains. In this experiment, we performed TL on a training dataset (target dataset) created from a different benchmark dataset than the dataset used in the pretext task (pretext dataset). Figure 7 shows the results of training on the target dataset, where the horizontal axis is labeled “pretext dataset → target dataset”. When transferred from HASC, SimCLR (seg) + SDFD showed a relatively high f1-score. In particular, when transferred to UCI-HAR, SimCLR (seg) + SDFD achieved 91.82%. When transferred from USC-HAD to HASC, SDFD had the highest f1-score of 66.01%. In the results of TL for different domains from USC-HAD to HASC, the SD-based methods showed a higher f1-score, while methods using SimCLR showed a lower score. The number of subjects in the USC-HAD training dataset was small compared to HASC and Wisdm. Therefore, we considered the lower f1-score of the SimCLR-based method caused by the smaller total number of training data, which prevented the model from obtaining a generalized feature representation. When the size of the dataset was not large enough during the pretext task, we assumed that the SDFD-based method was an effective approach to obtain generalized feature representations. In addition, since HASC, Wisdm, and UCI-HAR all have different sampling frequencies, we considered SimCLR (seg) + SDFD to be robust to different sampling frequencies when there was a sufficient amount of training data for the pretext task.

## 5. Conclusions

In this study, we tackled developing a new unsupervised representation learning method using segment information for HAR. We explained the measurement methods for sensor data to construct a HAR dataset. We introduced segment data, which are measurement data containing only one activity without annotating an activity label. The dataset used in our method consisted of instance data and segment labels, which were pseudo-labels assigned to the segment data. We proposed the new unsupervised learning method SimCLR (seg) + SDFD by combining SimCLR (seg) and SDFD. To investigate the effectiveness of the feature representation obtained by the proposed method, we conducted experiments by training only a linear layer attached at the end of an encoder with fixed encoder weights as TL. The experimental results showed that our proposed segment-based SimCLR methods obtained more-generalized feature representations with a higher f1-score than the regular SimCLR in HAR. We applied TL to different domains and found that the proposed SimCLR (seg) + SDFD performed robustly with respect to the sampling frequency of the sensor data. From these experiments, we suggest that the SimCLR-based method requires more training data than SDFD for the pretext task, but is effective when there are sufficient unlabeled data.

However, there are several limitations to consider. Firstly, this study used only accelerometer data for the datasets and did not discuss the quantity or other sensor data such as gyroscope or magnetometer data. Therefore, it can be expected that the accuracy of the proposed method will be improved by multimodalization. Secondly, this study only used a dataset with basic activities. It is expected that more-generalized feature representations could be obtained by using a dataset with a wider range of activities. Third, this study did not use a method to split the accelerometer data containing multiple activities into segment data. As future work, we will introduce multimodalization in the proposed method, using datasets with a wider range of activities and enhancing a framework that includes a method for splitting accelerometer data with multiple activities into segment data. These approaches will lead to more-accurate and -effective HAR systems.

## Figures and Tables

**Figure 1 sensors-23-08449-f001:**
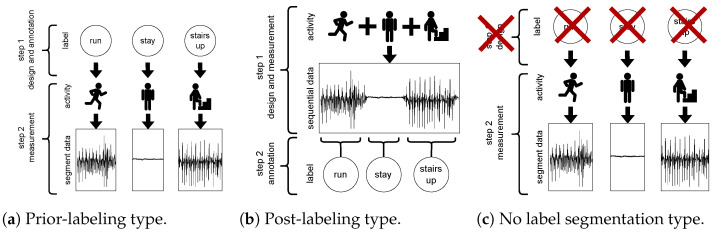
Data collection methods used in sensor-based human activity recognition (HAR).

**Figure 2 sensors-23-08449-f002:**
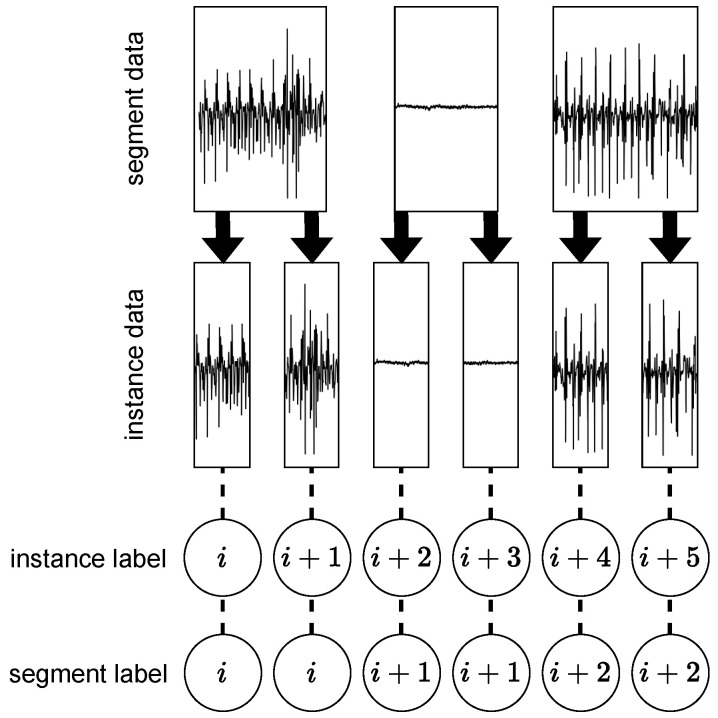
The difference in assigning pseudo-labels used in instance discrimination (ID) and segment discrimination (SD). An instance label (upper) is assigned to the input data in ID. A segment label (lower) is assigned to the segment data in the proposed SD. Therefore, the same segment label can be assigned to different input data.

**Figure 3 sensors-23-08449-f003:**
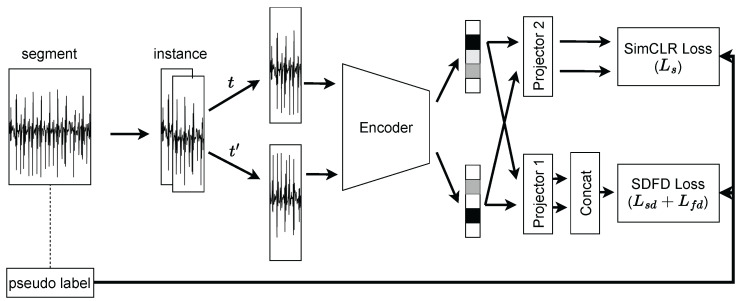
Overview of the proposed unsupervised learning method. We refer to the proposed segment-based SimCLR with segment discrimination and feature decorrelation as SimCLR (seg) + SDFD. For the model input, instance data are created from segment data upon which the model is then trained using the proposed loss function.

**Figure 4 sensors-23-08449-f004:**
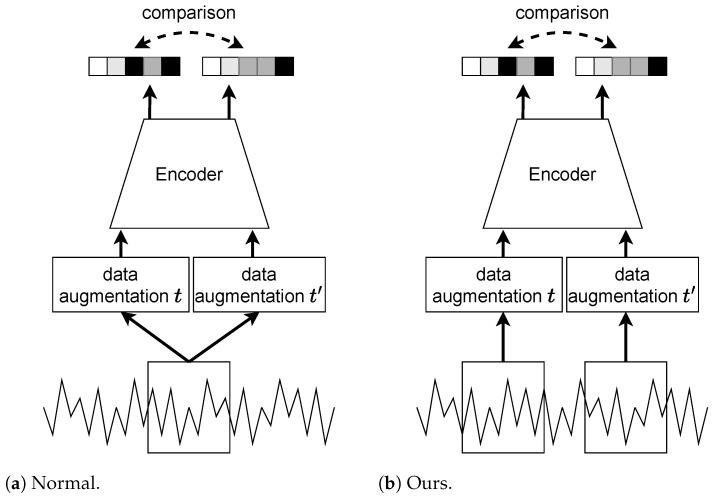
Difference in how positive pairs are created as follows: (**a**) Normal SimCLR creates positive pairs on an instance. (**b**) Segment-based SimCLR creates positive pairs from a segment.

**Figure 5 sensors-23-08449-f005:**
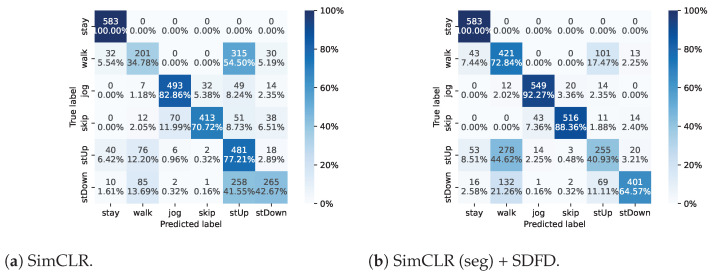
Confusion matrix with the number of instances and percentages. The axes represent the activities included in HASC. In particular, “stUp” represents “stair up”, and “stDown” represents “stair down”.

**Figure 6 sensors-23-08449-f006:**
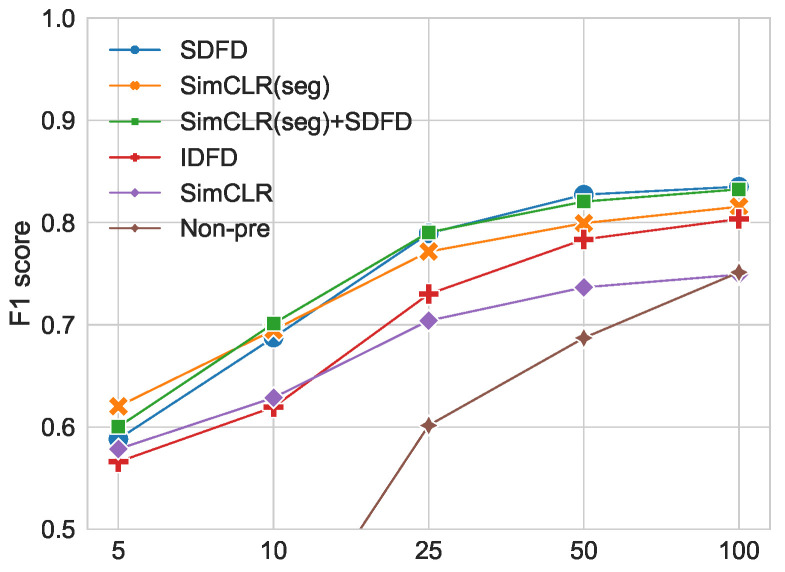
The number of labeled data and the f1-score of each model. The horizontal axis indicates the number of instances.

**Figure 7 sensors-23-08449-f007:**
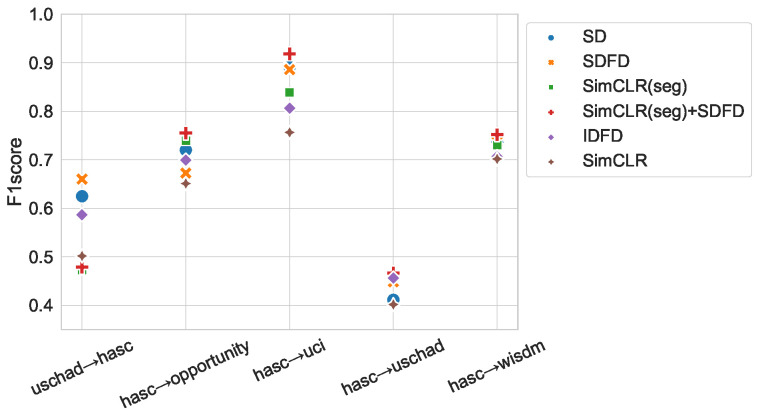
Transfer performance to different datasets. The x-axis indicates “pretext dataset → target dataset”.

**Table 1 sensors-23-08449-t001:** The number of subjects in each dataset. HASC [40] used a limited number of individuals.

Dataset	Training	Validation	Testing
HASC [40]	80	20	30
Wisdm [38]	21	6	9
Opportunity [42]	2	1	1
UCI-HAR [48]	19	5	6
USC-HAD [39]	8	2	4

**Table 2 sensors-23-08449-t002:** Average f1-score of each dataset in the same-domain transfer learning (TL). The encoder weights are fixed therein, and the number of instances is N=10. The bold font indicates the highest score, while the underlined font is the second-highest among the datasets used.

	HASC	Wisdm	UCI-HAR
Non-pre	0.3714	0.7286	**0.9397**
BYOL [34]	0.6072	0.5676	0.7198
SimCLR [18]	0.6287	0.6717	0.8492
ID [33]	0.2354	0.6661	0.8926
IDFD [32]	0.6195	0.6962	0.8986
SimCLR (seg) + SDFD (ours)	**0.7012**	**0.7636**	0.9108

**Table 3 sensors-23-08449-t003:** Ablation study. The bold and underlined fonts indicate the highest and second-highest f1-score, respectively.

	HASC	Wisdm	UCI-HAR
IDFD [32]	0.6195	0.6962	0.8986
SDFD	0.6875	**0.7980**	0.9047
SimCLR [18]	0.6287	0.6717	0.8492
SimCLR (seg)	0.6948	0.7679	0.8986
SimCLR (seg) + SDFD	**0.7012**	0.7636	**0.9108**

**Table 4 sensors-23-08449-t004:** f1-scores by activity for the HASC when TL with 10-shots per class. The bolded font indicates the most-accurate values.

	Stay	Walk	Jog	Skip	stUp	stDown
IDFD [32]	0.8970	0.3520	0.7752	0.6862	0.5017	0.5048
SimCLR [18]	**0.8992**	0.3916	0.7682	0.7092	0.4895	0.5147
SDFD	0.8977	0.4470	0.8548	0.8183	0.5170	0.5905
SimCLR (seg)	0.8756	0.4942	0.8525	0.8213	**0.5432**	0.5819
SimCLR (seg) + SDFD	0.8857	**0.5143**	**0.8738**	**0.8268**	0.4827	**0.6241**

## Data Availability

Not applicable.

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
