# Peer review of "Segment-Based Unsupervised Learning Method in Sensor-Based Human Activity Recognition"

_sensors, 2023, doi:10.3390/s23208449_

Round 1

Reviewer 1 Report

Title: Segment-based Unsupervised Learning Method In Sensor Based Human Activity Recognition

The research work reported is interesting. There, however, are some concerns that the authors need to address (major revision):

Major Concerns for Motivations & Contributions:

1.     The authors should highlight the manuscript’ motivations. What problems did the previous works exist? How to solve these problems? The authors may consider analyzing the problems of the previous works and how to address these problems in the manuscript. Please explain that.

2.     The authors must clearly explain the difference(s) between the proposed method and similar works in the introduction. The authors should highlight the manuscript’s innovations and contributions.

Experimental analysis and discussion:

1.     The experimental details are missing in the manuscript, e.g., the hyperparameters of the proposed method.

2.     The some crucial hyperparameters should be explained in the manuscript, e.g., why did the authors set λ1 = λ2 = λ3 = 1 in Eq. (5)?

3.     Ablation study is missing in the manuscript.

4.     The authors should compare the proposed method with some state-of-the-art algorithms, including representation learning algorithms, e.g., DCRLS (see 10.1109/TETCI.2023.3304948 and T-Loss (see 10.5555/3454287.3454705), and some state-of-the-art HAR algorithms such as perceptive extraction network and Multihead Convolutional Attention.

5.     Please consider providing the confusion matrices in the manuscript.

6.     Computational complexity should be analyzed and compared in the manuscript.

7.     Could you tell me the limitations of the proposed method? How will you solve them? Please add this part to the manuscript.

Reviewer 2 Report

First of all, I must draw your great attention to this literature from Germany by all means:

1. CSL-SHARE (A Multimodal Wearable Sensor-Based Human Activity Dataset) proposed and practiced several years prior to your work for "defining a data collection method that sensor data measured only one activity is named segment data" which you mentioned to be one of your key contributions. It provided a MULTIMODAL dataset with various types of wearable sensors (ACC X2, Gyro X2, EMG X4, Goniometer, Microphone...) with detailed SEGMENTED information. But it neatly solved the segmentation problem:

2. This work also proposed an easy-to-realize semi-automatic activity segmentation (and annotation) method DURING the data acquisition and did not need to label the data after the recording, which typically solved your described problem of "Annotators need to manually annotate activity labels for each measurement data which requires more significant human resources and time."

Since the a prior-labels were set in a protocol, subjects just needed to press the pushbutton, coinciding with your idea (the exact practice is different), but it predates you by several years—the first published literature on such appeared in 2018 (https://doi.org/ 10.5220/0006732902620268), and in 2019 there was A Wearable Real-Time Human Activity Recognition System using Biosensors Integrated into a Knee Bandage (Best Paper).CSL-SHARE is their follow-up, large-scale, 17-channel, involving 20 subjects and 400 hours of acquisition.

3. This dataset has been used for dozens of research publications as a useful base for various segment-based and non-segment-based HAR research works, which proves its annotation method's robustness and the acquired signal's quality.

Even though the above-mentioned work does not involve unsupervised learning, its collection ideas, dataset composition (segmented to "single activities") and usage fully conform to your concept.

Moreover, for segmentation of single activity on continuous signals (without having to care about their labels), DL of ML is actually not always a Must. On continuously recorded HAR datasets, Purely Statistical Models can also work efficiently, with minimal computational cost and surprisingly good results. In this regard, two models proposed in Portugal last year should be on your radar:

1. Feature-Based Information Retrieval of Multimodal Biosignals with a Self-Similarity Matrix: Focus on Automatic Segmentation excels at automatic segmentation of HAR datasets (Figure 6 ). Compared to your work's definition of "single activity" (which, by the way, I have something to say about later), this approach focuses on higher-order degrees of scope: when choosing different shifting window and kernel lengths, automatic segmentation can be accurate down to (1) block activity transitions, for example, standing->walking->sitting; (2) fine-grained activity switching, e.g., the above "walking" can be further divided into walking straight forward->walking upstairs->walking downstairs; (3) finer-grained gait segmentation, e.g., for each gait.

2. The query-based TSSEARCH method can also accurately segment a single activity from a dataset in a pre-specified series of "query sequences" (Figure 3). And DTW endows the method with activity morphology and length insensitivity: the model sequence can be a single gait or three steps, and the step to be examined can be slower or faster than the model pattern.

I am not denying the contribution of your work, but all the above-mentioned examples show a massive neglect of up-to-date closely-relevant literature that would make the manuscript seem arrogant to boast about its "new" idea. The fact is that many scientists have long been aware of the issues and ideas you have raised and made much effort to practice, improve, refine, SIMPLIFY, and even generalize (e.g., for all kinds of biosignals) them.

---------------

You mentioned SINGLE ACTIVITIES for segmenting, but in reality, it needs to be defined on a case-by-case basis.

First of all, brain activity is actually human activity, right? Of course, this is not something you need to explain because HAR itself is for kinematic situations. But what is a single activity? In the case of walking, does completing a walk (no matter how many steps) to change to a pre-defined "other" activity count as a single activity, or does a single step/stride/gait count as a single activity, or is it a stance-swing cycle of one leg (left/right)?

While reading the whole manuscript, it's unclear to me what exactly you're talking about when you say single activities. Then it is also unclear what "segments" you have segmented.

Since the single activity is at the core of the "segment-based" basis of this article, it needs to be taken seriously. Referring to 1.1 of Biosignal Processing and Activity Modeling for Multimodal Human Activity Recognition is highly recommended. The analogy between your single activities and the single motions in the literature, including locomotive ones and static ones, can be made. In addition, How Long Are Various Types of Daily Activities analyzed details of single motions. It proved that healthy adults' 22 single motions (walking forward/upstairs/downstairs/curvely/v-cut, jogging, one-leg jumping, two-leg jumping, standing up, sitting down, lateral shuffling...) durate all within 1-2 seconds and are normally distributed among the population. This could be borrowed into your definition. Again, please take this seriously.

------------

Regarding writing, there are many mistakes. Here are a few examples:

decision making (decision-making)

Figure 3 showS

each datasets (no "s") several times

We compared THE proposed method

split (no "ed")

Table 1 shows S

aN unsupervised

A partially ... setup

The above are just some examples. I deeply feel that the first half and the second half of the article were written separately by different people (that is, the seniors wrote the introduction, background, etc.; while the students wrote the experiment and analysis) because there is an obvious difference in writing quality. But eventually, no one did a rigorous review, cross-check, and unification. I am often interrupted while reading it. From Chapter 4 onwards, I can mark a dozen or even scores of grammatical errors on each page.

I also have some concerns on the machine learning part, e.g., no SOTA comparison, no confusion matrices, and no analysis of the recognition performance of which activities is good or poor in the proposed unsupervised model, among others, but I think the writing quality of the article is really poor, especially for the ML. So even enhancing the ML part cannot make this current manuscript publishable. I do not simply give "Reject", but please at least improve the manuscript to the level of reviewable, and we will discuss it later.

See above.

Round 2

Reviewer 1 Report

Thanks to the authors for answering my concerns. The authors had addressed my concerns. Thus, I recommend the journal accept the current manuscript.

Author Response

We thank you for reviewing our paper. We are pleased with your recommendation.

Reviewer 2 Report

The manuscript has been greatly improved by the author's revisions, of which I am positive. The author accepted the suggestions and practiced the changes in wording so that it did not give rise to more exaggerated claims. Things such as SOTA comparisons and confusion matrices were also formulated.

The revised version is commendable, but there are still some problems that do not meet the criteria for publication in the reputable journal Sensors. Besides, many concerns have arisen as a result of the content additions, some of which are critical errors.

1. Since the authors did not mention any about deep learning in the title or abstract, it is hard to understand why in L72-84 for sensor-based HAR, only DL (NN, Transformer, etc.) and related models are presented. This is another reflection of the authors' weak recherche on the literature. In fact, for sensor-based HAR, many non-deep models are equally valid. Even though the most shallow ML models such as KNN, DT, RF, SVM can indeed be skimmed over with one introduction sentence, the hidden Markov Model (HMM) has been reported in many works to achieve equal to or even exceeding performance than DLs, due to its inherent sequential modelling ability to efficiently and logically model HAR time series. HMMs also outperform DLs in terms of excellent interpretability, expendability, and generalization, and can be sufficiently applied to real-time HAR with low computational cost (On a Real Real-Time Wearable Human Activity Recognition System).

Furthermore, the retrospect of sensor-based HAR in Chapter 2 is limited to the software level (AI) and ignores the hardware. The two aspects are complementary. Stay tuned for this year's up-to-date summary: Sensor-based human activity and behavior research.

2. Eq. 4: why abbreviated division with a "/" if you are using a formula structure rather than an in-line formula?

3. some subsections' headings are unacceptable, e.g. 3.3.1 – 3.3.4. Readers don't know their meaning without reading the whole article through. The catalog (Chapter, Section, Subsection) is meant to be indexed for the reader: do you want the reader to look at your section and subsection names and be unclear about what they describe and unable to find the content they are interested in?

4. abbreviations that have not been given the full name are unacceptable, even if you are tacitly assuming that the reader in relevant fields understands what you are talking about.

5. abbreviations appearing in the section/subsection/table/figure headings need to be given the full name when they first appear, even if the full name has already been given earlier in the textual content.

6. sliding window and window size are mentioned in L152, L240, etc. In fact, the choice of window size is a topic that needs to be investigated, but the authors have taken a default approach in the text to avoid investigating this issue.  The latest How Long Are Various Types of Daily Activities studying on the duration of 22 human activities can give a kinematic indicator to this aspect: human's daily individual activities (e.g., skip, stup, stdown, walk (a gait), jogging (a gait) in your manuscript) fall within the 1-2 second range and are normally distributed in the population. Combined with the sampling rates of different sensor devices, the above research facts provide a more stable numerical reference and tuning interval for window size selection.

7. main text: "Table 2 shows the average f1 scores". title of Table 2: "Accuracy of each dataset". Do the authors understand that F1 score and accuracy are two different evaluation metrics? Even in Japanese, F値 and 正確度/精度 are different words.

8. the coloring of the confusion matrices (CM) in Figure 5 is wrong. This is a common newcomer's mistake. The color should represent the percentage of accuracy of the cell it is in (not how many instances). As a simple example, if there is a dataset containing two classes: walk and jog. walk contains 200 labels and jog contains 100. And some classifier identifies them exactly right—what CM will be obtained by applying your scheme? It's predictable that the background color of jog is half as light as that of walk, right? And the color has an opinion in the CM: the correct proportion. So you should count percentage to determine color, not instance numbers. 

8.1 Replace instance numbers in CM with percentages. It is more recommended to show both

8.2 Determine background color based on percentage

Feature Space Reduction for Multimodal Human Activity Recognition, Figure 3: A Wearable Real-time Human Activity Recognition System using Biosensors Integrated into a Knee Bandage, Figure 4; and Biosignal processing and activity modeling for multimodal human activity recognition, Figures 5.24-5.32 give examples of plotting CM using only percentages (case numbers are omitted because of the large number of activity classes). For your Figure 5, given that there are only six activity classes, it could be shown in two lines per cell, one for case number and one for percentage.

By the way, some literature mentioned above are precursors to your work that I mentioned in Round 1, but they were omitted.

An image from a piece of literature about to be published is given attached that can help you design the CM plot.

In terms of investigation of earlier work, design of confusion matrices, misuse of evaluation metrics, etc., I am not just "reviewing" the authors' manuscript, but literally "teaching" it. Of course, I am happy to do this in order to protect academic comers, but I am not sure that every reviewer will be so patient. I hope you will further optimize the article.

See above.

Round 3

Reviewer 2 Report

The author carefully revised the article to address the concerns I raised. I argue to receive the manuscript.